# Impact of Environment on Pain among the Working Poor: Making Use of Random Forest-Based Stratification Tool to Study the Socioecology of Pain Interference

**DOI:** 10.3390/ijerph21020179

**Published:** 2024-02-05

**Authors:** Eman Leung, Albert Lee, Yilin Liu, Chi-Tim Hung, Ning Fan, Sam C. C. Ching, Hilary Yee, Yinan He, Richard Xu, Hector Wing Hong Tsang, Jingjing Guan

**Affiliations:** 1JC School of Public Health and Primary Care, The Chinese University of Hong Kong, Hong Kong SAR, China; 1155189592@link.cuhk.edu.hk (Y.L.); cthung@cuhk.edu.hk (C.-T.H.); chuncheungching@cuhk.edu.hk (S.C.C.C.); yinanhe@cuhk.edu.hk (Y.H.); jingjingguan@cuhk.edu.hk (J.G.); 2Department of Rehabilitation Science, Hong Kong Polytechnic University, Hong Kong SAR, China; richard.xu@polyu.edu.hk (R.X.); hector.tsang@polyu.edu.hk (H.W.H.T.); 3Centre for Health Education and Health Promotion, JC School of Public Health and Primary Care, The Chinese University of Hong Kong, Hong Kong SAR, China; 4Hong Kong Health Education and Health Promotion Foundation, Hong Kong SAR, China; 5Centre for Health Systems and Policy Research, The Chinese University of Hong Kong, Hong Kong SAR, China; 6Health in Action Limited, Hong Kong SAR, China; fanning@hia.org.hk; 7Faculty of Medicine and Health, The University of Sydney, Sydney 2006, Australia; hyee0574@uni.sydney.edu.au; 8Epitelligence, Hong Kong SAR, China

**Keywords:** pain interference, working poor, built environment, machine learning

## Abstract

Pain interferes with one’s work and social life and, at a personal level, daily activities, mood, and sleep quality. However, little research has been conducted on pain interference and its socioecological determinants among the working poor. Noting the clinical/policy decision needs and the technical challenges of isolating the intricately interrelated socioecological factors’ unique contributions to pain interference and quantifying the relative contributions of each factor in an interpretable manner to inform clinical and policy decision-making, we deployed a novel random forest algorithm to model and quantify the unique contribution of a diverse ensemble of environmental, sociodemographic, and clinical factors to pain interference. Our analyses revealed that features representing the internal built environment of the working poor, such as the size of the living space, air quality, access to light, architectural design conducive to social connection, and age of the building, were assigned greater statistical importance than other more commonly examined predisposing factors for pain interference, such as age, occupation, the severity and locations of pain, BMI, serum blood sugar, and blood pressure. The findings were discussed in the context of their benefit in informing community pain screening to target residential areas whose built environment contributed most to pain interference and informing the design of intervention programs to minimize pain interference among those who suffered from chronic pain and showed specific characteristics. The findings support the call for good architecture to provide the spirit and value of buildings in city development.

## 1. Introduction

The estimation of pain prevalence adjusted for age and sex is found to be 27.5% across 52 countries [1], which is even more prevalent among low-income countries according to a recent meta-analysis, reporting that 33% of their adult populations and 56% of their elderly populations suffered from chronic pain [2]. In addition, even among studies published from populations of high-income countries, respondents with lower income suffered from chronic pain, and they were more likely to report pain interfering with different aspects of their lives [3,4,5]. However, those with lower income also showed greater variability in whether, and to what extent, pain interference was experienced compared with those with higher income [6]. Yet, systematic research into what factors are responsible for such variability is lacking, despite its potential role in identifying targets for interventions that can yield the greatest return on investment.

A recent update by Trachsel et al. has highlighted that the activation of peripheral and central sensitization pathways involves several mechanisms sensitizing the peripheral nociceptors and altering the spinal dorsal horn neurons and central nervous system (CNS) brain areas to trigger a pathogenetic cascade that ends with the development of chronic pain [7]. Again, pieces of evidence suggest the paramount role of the environment (i.e., epigenetics) and genetics [7]. Pain interference [8], commonly operationalized by averaging the rating of pain interference across areas such as daily activities, professional duties, social relationships, mood, and sleeping quality [9,10], is predisposed by the demographical, clinical, psychological, and behavioral characteristics, e.g., the intensity and locations of pain, and the chronic conditions to which the pain in question was linked, which have been associated with the averaged rating of pain interference [11,12]. Interference by pain has been shown to link with demographic factors and psycho-social factors [13], pain-related catastrophizing [14,15], and pain acceptance [16,17], and behavioral and bio-medical factors, such as obesity [11] (for linkage between pain interference and obesity-related arthrometries and biometrics such as BMI, blood pressure, and blood glucose [18]) and tobacco use [19].

A recent study has shown that people with a lower income experience greater pain than those with a higher one, highlighting the role of psychosocial factors in physical pain [20]. In addition, studies also suggest that those with lower income experience relative deprivation [21] and social subordination [22], which may result in depression, anxiety, distress and other negative emotions such as anger, stress, and resentment—which has been linked to greater levels of pain [23] and pain interference [24] perceived compared to those with better psychological well-being. 

Built-in environmental-level factors also hold significant importance for pain and pain interference, even though they are often overlooked. Patterns of co-residence [25] and housing environment [26] have been linked to pain interference in daily activities. A systematic review has found a relationship between numerous aspects of healthy building determinants and back and neck pain [27]. A study has shown that the presence of nature in the urban residential environment reduces pain intensity among community-dwelling middle-aged and older adults living with chronic pain by buffering the impact of pain catastrophizing and rumination. [28]. A built environment characterized by sufficient lighting [29,30], large space [31], good indoor air quality [32], and accessibility to activity [26] can effectively improve pain and pain interference with sleeping quality, vitality, and mood. Quality of built environment, such as construction type [33], age [34], design [35] and size of living area [36], had been linked to health. Moreover, these built-environmental characteristics can affect health through biological connectivity, such as indoor mold [37], dust [37], and microorganisms [38]. For instance, building design can impact microbiome load, which is associated with the risk of disease [39]. Notwithstanding the knowledge about the impacts and mechanisms of built environments on health, the association between housing environments and pain interference remains inadequately explored, with little attention paid to investigating this relationship in conjunction with individual-level factors, even though the effects of built environments on its residents are usually compounded by the residents’ age, sociodemographic, and personal behavior [40]. Furthermore, the linear model-driven approach commonly applied to analyze factors that predispose to pain interference is unable to tease apart the unique contribution of different individual- and environmental-level factors that are intricately interrelated.

Isolated studies have also linked pain interference with individual aspects of one’s living environment, such as small living areas [26] (operationalized here as the number of non-functional rooms being two or less), poor air quality [26] (operationalized here as the buildings lacking re-entrant bay [41]), poor access to light [42] (cardinal orientation, and specifically in the local setting, those residential buildings facing south have better access to light [43,44]; and the number of light wells in the building [45]), social connectivity [5,27,46] (operationalized here as the buildings’ numbers of corridors and lifts [47,48,49,50]), and the age of the building.

Hence, to model and risk-score the unique and combined contribution of a diverse ensemble of environmental, sociodemographic, and clinical factors to pain interference among the working poor, a novel application of the random forest algorithm was reported here. The random forest is a commonly used machine learning algorithm to combine the output of multiple decision trees to enhance their performance, lower the requirement for sample sizes, and maintain the interpretability of the findings. Here, to further enhance random forest interpretability and our findings’ implementability in the context of community-based screening, a risk-scoring system was subsequently developed from the result of the random forest model with validation. Details are described under Section 2.3.

## 2. Method

### 2.1. Participants

A retrospective observational study was conducted among members of a community service of a Non-Government Organization (NGO) providing self-management interventions that focus on pain, physical therapy, and mind–body wellness counselling to residents of a Hong Kong district living in working poverty, as defined by having spent at least 27 weeks in a year working or looking for employment, yet household income falls below the poverty line [51], which is 50% of the median monthly household income before policy intervention [52].

Full data were ascertained from 568 members of the community service who were (1) deemed to be suffering from pain by healthcare professionals of the community service, (2) had received between June 2021 and October 2022 assessments on pain, pain interferences, BMI, blood pressure, and blood glucose by healthcare professionals of the studied community service, and (3) reported their demographics and smoking and drinking habits. Of the 568 members, 335 were also residents of public housing and were included in the study cohort.

Members of the community service consented to the data collection for quality improvement and dissemination purposes. (ICPC-2). Ethics approval for secondary data analysis was obtained from the Joint Chinese University of Hong Kong-New Territories East Cluster Clinical Research Ethics Committee. The study subjects were clients of the studied NGO who gave consent for the analysis and reporting of their data in an anonymized aggregated form for the purposes of service quality improvement, education and research. The studied NGO has consented to our analysis and reporting of the data under their custodianship under strict confidentiality agreements safeguarding the anonymity of their clients while performing service quality improvement, education and research. This study has been approved by the Chinese University of Hong Kong Research Committee—the Human Subjects Ethics Sub-Committee.

### 2.2. Predictive Features and Pain Interference Outcome

The predictive features and outcomes of the current study were extracted from two main sources of data, including those that were available in the public domain (i.e., the Housing Authority’s database on the architectural elements of public housing buildings in the studied district) and those that were collected by the studied community service, such as data on its clients’ demographics, anthropometric and biometric measures (i.e., BMI, blood pressure, and blood glucose), and the presence, location, pain severity and interference assessed by the Brief Pain Inventory (BPI) [53]. Specifically, BPI is a validated tool designed for assessment of the nature of pain and its interference in the following areas: daily activities, professional duties, social relationships, mood, and sleeping quality. While most studies operationalized pain interference by averaging the mean scores across the five above-mentioned areas as a single index, this study examined the presence of pain interference in each area as separate binary outcomes and the number of areas suffering from pain interference as ordinal counts to reflect the severity and pervasiveness of pain interference in the lives of the study participants. The PBI was administered during the same session as when biometrics were assessed.

The literature has linked abnormal BMI, blood glucose, and blood pressure values to pain interference. However, those in our sample who reported abnormal BMI (23.8%, 32.4% and 9.5% fell into the categories of pre-obesity, mild obesity, and severe obesity, respectively) did not always show abnormal levels of blood pressure (21.4%) or blood glucose (16.2%), despite the relationships among the three. The discrepancy may be attributable to the use of medication for managing blood pressure and glucose, which had not been collected in the current study.

The following features were adjusted in the random forest model due to their potential contributions to pain and/or pain interference: residential population density [49,54], number of distinct households in the building (operationalized as the number of flats in the building—for the relationship between social connectivity and the number of district household in the building [55,56], living in subdivided flats (reflecting adverse living conditions in the local context; for the relationship between sub-divided flats and physical and mental health) [57,58], behavioral risk factors such as drinking [59] and smoking [24], whether or not the participants had been assigned any pain management care plan (whereby such assignment decision may be associated with differences in participants’ conditions that were not captured in the community service’s database), the severity of pain experienced [12,13], and whether the pain assessment was performed during wave five of the COVID-19 pandemic or during the period between wave four and five when COVID-19 was laid dormant (as pain and its interference is affected by COVID-19 [60,61] as the different in the social and public health measures against COVID-19 during wave five and the period leading up to it may affect participants’ professional and daily activities and how pain might affect them).

### 2.3. Analytic Models: Model-Based Feature Selections and Scoring

The current study applied a random forest-based algorithm [62,63] to: (1) model the study outcomes; (2) select features by integrating individual decision trees’ rank-ordering of features, which are based on each feature’s unique and combined contribution to the study outcomes; and (3) optimize the cutoff values that define the response levels of selected features and the weights to assign to the corresponding response levels of the selected features. Below, the process with which random forest selects and rank order features for splitting the sample will first be described, followed by a justification for our using the random forest-based algorithm here instead of other more traditional analytics. For example, the data preparation process for constructing our pool of predictive features is minimized with the algorithm presented here. Finally, the criteria by which the performance of our models will be evaluated are laid out.

Specifically, features were selected for the final scoring system by a sequence of modeling steps. Please refer to Figure 1 for a schematic illustration and the corresponding pseudo-code for a high-level comparison of the process of feature selection and splitting between decision trees and random forests when performing classification tasks.

Firstly, multiple decision tree models were performed (when the number of models reached 500 or the performance of models converged at maximum, whichever came first). With a recursive splitting process, each decision tree compares all predictive features’ unique and combined contributions to the studied outcome with each other and then selects predictive features in an order that reflects the magnitude of their respective contributions to the studied outcome. The decision tree’s recursive splitting process stops when additional splits on previously unselected features fail to differentiate the remaining participants with respect to the studied outcome.

Secondly, the results of individual decision trees were combined via an ensembling process, whereby features selected by individual trees (i.e., subsets of features drawn from the original feature pool) were integrated and bootstrapped randomly to create samples of data from which the unique statistical importance of each feature was estimated according to the corresponding drop in predictive accuracy when one feature’s value is replaced with its random permutation value given the supervisory outcome.

There are a number of advantages that the random forest algorithm reported here has over individual decision trees or even random forest algorithms more generally. A non-regression-based decision tree can model the unique and combined contributions of intricately interrelated predictors, which generally violates the assumption of traditional linear models with high performance while achieving greater interpretability compared to other machine learning and deep learning models. Nevertheless, while the random forest model maximizes the differentiability of individual decision trees to enhance its accuracy in model prediction and variable ranking and optimize the estimation of classification probabilities with ensemble logistics functions [64,65,66,67,68,69], the novel random forest algorithm deployed here yields more interpretable results than random forest models in general or individual decision trees due to its enabling the assignment of weights that maximize outcome differentiability to each response level of the selected features.

In addition, unlike traditional regression models or decision tree models, where extensive data preparation and feature transformation are required. The random forest-based algorithm deployed here can incorporate mixed data modes that include continuous, ordinal and categorical features. For example, continuous variables are first transformed into ordinal ones before assigning weights to each ordinal level according to the quantiles of the variable’s value (e.g., 0%, 5%, 20%, 80%, 95%, 100%). Consequently, every response level of each of the selected features was assigned a partial (marginal) weight, which is derived from the resulting coefficients through a two-step procedure. In the reference category, we assigned the smallest coefficient, and then the coefficients assigned to the non-reference response level were divided by the reference coefficients and rounded to the nearest integer. Notably, the level of a feature with lower importance may be assigned a greater weight than the level of another feature with higher importance. Hence, the model-based weight assignment further enhances the added interpretability that the common measures of statistical importance have brought to machine learning or deep learning models.

The validation dataset reported the model performance, and 95% confidence intervals (CI) were calculated by applying 100 boot-strapped samples [70]. The model’s discriminatory performance was considered poor if Area Under Curve < 0.70 [71], acceptable when AUC = 0.70 to < 0.80, excellent when AUC = 0.80–0.90, and outstanding when AUC > 0.90 [72].

## 3. Result

When applied to a feature pool consisting only of individual-level features, the random forest models supervised by mood, daily activities, sleeping quality, social life, and work performance yielded AUCs of 0.76, 0.78, 0.78, 0.79, and 0.84, respectively.

Table 1 lists and provides descriptive statistics for the predictive features and the model-supervising outcomes examined in the current study. As Table 1 shows, 77.8% of the study participants were female, aged 55.58 on average (SD = 12.1). The top three occupations were domestic helpers (37.1%), cleaning workers (25.9%), and catering staff (10.3%). Notably, 36.2%, 32.4%, and 29.5% of the participants reported pain in their knees, shoulders, and hands, respectively (pain can appear across multiple locations for any participant). The majority of the participants reported pain interference with their daily activities (72.9%), sleeping quality (50.0%), and work performance (46.7%), with 75.2% of the participants reporting at least two areas of pain interference (46.2% reporting three or more areas).

Table 2a shows the features selected and the order in which they were selected, by an individual-level random forest model supervised by binary outcomes of whether pain interference was present in areas such as mood, daily activities, sleeping quality, social life, and work performance. The summative score derived from aggregating the reverse-coded ranks of each feature with respect to the five outcomes assigned was tabulated (Table 2a).

In addition to age, occupation, and the intensity and location of pain, the BMI and blood glucose of those suffering from pain were consistently the top-ten features contributing to pain interference across all areas studied. In contrast, the rest of the selected features were found to have less consistency across the different areas of pain interference with abnormal blood glucose levels, and different pain positions ranked sixth and the rest of the top ten, respectively, when the summative score is concerned.

On the other hand, when applied to a feature pool consisting of both individual- and building-level features, the random forest models supervised by mood, daily activities, sleeping quality, social life, and work performance yielded AUCs of 0.96, 0.98, 0.94, 0.99, and 0.99, respectively.

Table 2b shows the features selected and the order in which they were selected by an individual- and building-level random forest model supervised by binary outcomes of whether pain interference was present in areas such as mood, daily activities, sleeping quality, social life, and work performance. Table 2b shows the summative score derived from aggregating the reverse-coded ranks of each feature with respect to the five outcomes assigned.

Features such as age, occupation, pain intensity and BMI were found to remain in the top ten in terms of their summative scores. With respect to the five studied areas of pain interference, the rankings of their importance were less consistent and lower (except age) compared to those received from individual-level-only models. Instead, the cardinal orientation of the building, the proportion of flats with less than three non-functional rooms, the building age, and the number of corridors and lifts were found to be the top-ten areas of importance according to the summative scores, and for most of the individual studied, these were areas of pain interference.

Table 3 shows the weights assigned to each level of every feature selected by the models whose supervisory outcomes were the total number of areas each member suffered from pain interference. What’s shown in Table 3 demonstrates how the random forest-based tool stratifies individual members of the working poor according to their respective risk of pain interference. The total number of areas inferred by pain was used as the supervisory outcome rather than the binary outcome of whether pain interference was present in individual areas. Table 3 reports the weights assigned to features selected by ordinal count-supervised random forest models when applied to a pool of individual-level features (left panel) and a pool of mixed individual- and building-level features (right panel).

Table 4a,b report the number of areas suffering from pain interference across the different ranges of values of the total score that was tallied from the weights assigned to individual-level features and individual- and building-level features, respectively.

As shown in Table 3, when applied to a feature pool consisting of only individual-level features, the ordinal count-supervised model (the left panel) and binary outcome-supervised models (as shown in Table 2a) assigned top-ten importance to similar sets of features, such as age, occupation, pain intensity, BMI, blood glucose, and locations of pain. In addition, the weights assigned to features selected by the individual-level model further enhanced the interpretability of the relative contributions of different response levels of a single feature and among all selected features. Among the working poor suffering from chronic pain aged between 44 and <73 and working as a clerk, reported pain interference in the largest number of areas (Table 3 left panel). The greater the number of areas of reported pain interference, the greater the pain intensity. Pain interference was also reported in a greater number of areas when the remaining top-ten individual-level features were present, i.e., BMI that was pre-obese or obese, abnormal blood glucose level, or pain located in the knee, lower back, or foot.

However, the weights assigned to the response levels corresponding to these features were found to be lower with modeling the ordinal count as an outcome with mixed building- and individual-level features (right-side panel of Table 3) than without building features (left-side panel of Table 3). While the rankings of pain intensity were found to be similar between the individual-level model and the mixed individual- and building-level model, the weights assigned to the response levels of pain intensity features in the mixed individual- and building-level model were found to be lower in value, thus contributing less to having multiple areas of pain interference. In addition, the same individual-level features can also be assigned a lower ranking, and their corresponding response level, the lesser weights (or different distributions of weights), in the mixed model, such as BMI, age, and occupation. Instead, the following building-level features’ response levels were assigned the greatest weights, contributing to the risk of having a greater number of areas interfered with by pain: cardinal orientation other than the south (south, southwest, and southwest) side, lower than 12.5% of flats having three or more non-functional rooms in the studied buildings, buildings having 564 or more flats, and years of building being 17 years or more.

## 4. Discussion

The study has revealed that the working poor’s internal built environment, such as the size of the living area (parameterized as the number of non-functional rooms in the flat), the building’s access to light, the quality of air, and the age of the building, were assigned as much (and sometimes higher, depending on the supervisory outcomes) statistical importance than other more commonly examined predisposing factors for pain interference, such as age, occupation, the severity and locations of pain, BMI, serum blood sugar, and blood pressure. It is of note that the effects of one’s internal built environment remain dominant even after the unique and combined effects on pain interference have been accounted for in our random forest model. The severity and locations of pain, BMI, serum blood sugar, and blood pressure are often studied in the literature in terms of their unique contribution to pain interference. However, the literature has rarely examined these factors systematically among the working poor. Nor were these factors examined in relation to the contribution of one’s age, occupations, and living environment. In fact, given that their effects on pain interference are multileveled and potentially non-monotonic, they had not been examined systematically in a single study with other more well-studied determinants.

Notably, the current study adds value to the literature on the social determinants of pain interference. Many studies have emphasized the association between age and pain interference [73,74]. Our study suggested that those aged between 44 and 73 suffered from the most pain interference, followed by those younger than 44, yet no significant effect was observed among those aged over 73. Hence, findings reported here is in congruent with that of other studies who found elders reported less pain interference with daily activities [75] and mood [74]. Yet, our findings are different that of Thomas et al., which indicated that the occurrence of pain interference with activities increased monotonically with age [73]. The potential reason for the difference could be that we examined the presence of pain interference in professional activities and social relationships, which might be less affected among those who were above 73 despite their working status. It has been shown that manual work is often associated with greater pain interference, with construction workers reporting the most interference in their daily activities, according to a local report [76]. Catering employees and grocery clerks were found to be more likely to report pain inference, and this would be due to no consecutive days of rest and stress at work [77]. Our findings also resonate with another recent study suggesting that workers engaging in physical activity related to transportation experienced less pain interference, whereas those performing household physical activity reported higher pain interference [78]. While our findings are consistent with the results reported in isolated studies, we add value to the body of literature by quantifying the relative contributions of different levels across all selected features to enable a comparison to be made with the intricately interrelated effects of all determinants adjusted and by examining these effects in a sample of working poor.

Significantly, our having identified that aspects of one’s built environment are linked to one’s experience of pain interference enable one to target key environmental factors prevention or mitigation, as brain imaging studies has shown with rodent pain models and among pain patients that environmental factors can reverse the changes in brain’s gray matter volume, white matter integrity, and epigenesis linked to the experience of pain’s impact [79,80]. In addition to brain imaging studies, our findings also resonate with other population studies’ findings in how environmental factors reverse pain-related outcome. For example, it has been shown that having nature nearby can buffer the relation between catastrophizing and pain intensity, as the nearby nature moderates the association between pain-related rumination and pain intensity but not the helplessness-pain intensity or the magnification-pain intensity associations [28]. Hence, healthcare professionals might look for community resources, such as nearby nature, for pain management [28]. In addition, a study in China has shown that people who lived in moderate and unfavorable environments had higher risks of arthritis in cross-sectional analysis and follow-up studies. An inferior living environment might promote the development of arthritis, so improvement of the living environment can be a good strategy for the primary prevention of arthritis [81]. Similarly, a study examined the context of environmental life spaces with chronic osteoarthritis pain among older African Americans aged 61–81 living in income-adjusted housing [26]. Their housing environments lacked age-friendly amenities, and specific enhancements to assist aging individuals with everyday function and reduce pain were identified [26]. This further illustrates the importance of understanding the impact of environments (macro, meso, and micro) on pain management [26].

One of the potential factors identified as being a key mediator of the effect of environmental factors on the experience of pain and its interference is one’s psychological well-being [82]. For example, a UK Biobank study (with over 150,000 participants) has shown that distinct neurobiological changes associated with specific psychiatric symptoms are linked to certain environmental factors [83]. In particular, the study found that air pollution and social deprivation of one’s living conditions were linked to affective symptoms and a concurrent shrinkage in the brain area that processes reward and heightened expression of stress-responding genes. On the other hand, the study also found that accessibility to nature can protect one against anxiety symptoms, with differences in the activity level of the brain regions necessary for emotional regulation mediating such protective effect.

On the other hand, the associations between built-environment quality and people’s mental health were previously not found to be strong in Hong Kong [84]. The impact of housing characteristics on mental health would be more direct in communities with relatively poor housing conditions, and the effect might be indirect in communities with relatively good housing conditions [84]. Hence, it is more important to improve the living environment of the working poor and hope to improve their mental health to avoid aggravation of their perception of pain. Notwithstanding the well-documented association between the built environment and pain, or health and mental health in general [27,84,85], there are not many studies examining the impact of the built environment on pain interference specifically. The effect of age of the buildings on pain interference had not been examined directly in the literature, and previous studies have suggested older buildings are usually found in improvised neighborhoods lacking well-organized facilities, and the inconveniences would cause the residents suffering from pain and limiting their socialization, thus further aggravating their perception of pain interference [5,86]. While density of residential population, social connectivity in residential building layout, and poor ventilation have separately been linked to poor health [87], and particularly the building’s poor ventilation has been linked to a higher risk of pain [32,88], their linkages to pain interference have never been studied. Nor have their collective effects embodied in particular public housing designs ever been explored in the context of pain interference. The prevalence is nevertheless highest among low-income populations, who were more likely to populate public housing compared to middle- and high-income populations.

In addition to isolating individual determents’ effects from an intricately related ensemble of factors with a novel machine learning model, we have applied an interpretable approach to quantifying the relative weights of the unique and combined contribution of each factor selected by random forest models to enable decision-makers to target the allocation of resources at the frontline and the development of policies to address long-term needs. For example, when cross-referencing the model-based assignment of weights to the mixed building- and individual-level features reported in the right column of Table 2 with the scoring system shown in Table 3. On the one hand, the decision-makers can narrow the perimeter of the community screening for pain and its interference to older buildings with smaller living areas that were built with the construction models that received the greater weights (or, more broadly speaking, those with poor ventilation, high connectivity, and a social mix) and target building residents who are working poor and age between 44 and 73 with clerical or catering jobs for more in-depth assessment and intervention. The decision-makers can also put in place occupational health policies targeting low-income middle-aged individuals working in specific jobs for pain management programs, as well as housing policies to mitigate the adverse effects of poor ventilation and high connectivity in current and future public housing.

In addition, the current study is in alignment with a small but emerging body of literature that classifies clinical outcomes by modeling the unique and combined contributions of an ensemble of comprehensive and intricately related features with a random frost algorithm. One example of using a random forest algorithm is to investigate important variables for patient safety, as researchers can predict accurate and stable relationships between variables, and healthcare quality knowledge, organizational factors, and top management objectives were found to play a crucial role in determining patient safety grades [89]. The algorithm can automatically handle interactions with accurate predictions even when a large number of variables are present. Random forest algorithms have built-in cross-validation capability to enable the independent variables to be ranked according to their association with the outcome variable, from most effective to least effective [90]. There are many variables associated with health outcomes in the real world of healthcare delivery, and it is not easy to predict the relationship between variables even with advanced statistics. The random forest algorithm is useful for the investigation of risk predictors of different health outcomes in a complex healthcare environment. On the other hand, the current study adds additional value to the literature with its novel application of the random forest algorithm to drive the development of a scoring system whose purpose is to identify in community settings the working poor at risk of pain interference across different areas.

The limitations of the current study include that its participants were recruited from only one NGO, and the sample size is not very large. The study subjects are from grass roots, mainly living in poor conditions, which fits the criteria of working poor. The random forest-based algorithm can select features by integrating individual decision trees’ rank-ordering of features based on each feature’s unique and combined contribution to the study outcomes and optimizing the cutoff values defining the response levels of selected features and assigning the weights to the corresponding response levels of the selected features. The method can accommodate the current sample size. It would further enrich the findings if future studies could recruit members from the local District Health Center (a hub for primary healthcare services), and the Center would conduct pain assessments into different categories of severity. Similar studies would be conducted in other districts, including those with better living conditions and higher socio-economic status. This would enable a more detailed comparative analysis of the impact of the environment on pain for people of different socio-economic statuses.

## 5. Conclusions

There is a benefit to informing community pain screening to target residential areas whose built environment contributed most to pain interference and informing the design of intervention programs to minimize pain interference among those suffering from chronic pain and showing specific characteristics. Pain interference among the working poor has a significant impact on their livelihood, such as decreased productivity, impaired job performance, and limitations on daily activities. The findings of this study can serve as a reminder for primary care physicians to obtain a detailed social history of their patients to identify those patients with high living environmental risks to assess pain and its interference for effective prevention. Early intervention would empower patients with pain management skills to minimize pain and suffering, prevent further deterioration, and restore functional activity as far as possible. The findings support the call for good architecture to provide spirit and value to buildings, which are the nodes of a city, as described by Woo and Ko in a chapter on city development in Hong Kong [91].

## Figures and Tables

**Figure 1 ijerph-21-00179-f001:**
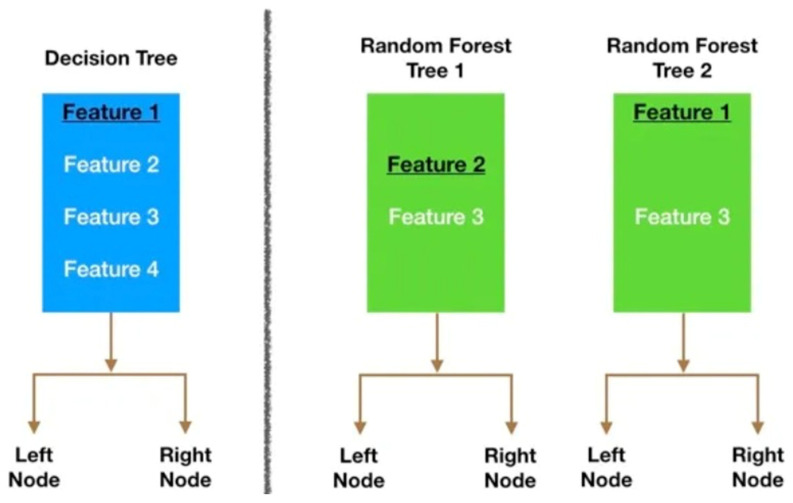
Schematic illustration comparing the selection and splitting mechanism between single decision tree and random forest. Note: The schematic represents the steps of random forest algorithms at a high-level: (1) Take the original dataset and create N subsample of size n, with n smaller than the size of the original dataset. (2) Train a decision tree with each of the N subsamples’ datasets as input. But when doing a node split, do not explore all features in the dataset. Randomly select a smaller number of features (M) from all the features in the training set. Then, pick the best split using impurity measures, like Gini Impurity or Entropy. (3) Aggregate the results of the individual decision trees into a single output. (4) Preform the classification task by taking a majority vote across all trees, for each observation.

**Table 1 ijerph-21-00179-t001:** Descriptive statistics of patients’ features and distribution of reported areas of pain interference.

Pain Interference Outcome/Feature		Prevalence
Mood	No	58.1%
	Yes	41.9%
Daily Activity	No	27.1%
	Yes	72.9%
Sleeping Quality	No	50.0%
	Yes	50.0%
Social Relationships	No	74.8%
	Yes	25.2%
Work Performance	No	53.3%
	Yes	46.7%
The ordinal counts of how many areas suffered from pain interference	0–1 item	24.8%
	2 items	29.0%
	3 items	31.0%
	4+ items	15.2%
**Predictive Features**		
Age	Mean (SD)	55.58 (12.08)
Sex	F	77.8%
	M	22.2%
Main Occupation	Catering Staff	10.3%
	Cleaning Worker	25.9%
	Clerk	6.0%
	Driver	1.1%
	Domestic Helper	37.1%
	Sales	4.3%
	Security Guard/Watchman	10.3%
	Technician	2.5%
	Construction/Manual Worker	2.5%
Blood Pressure (WHO standard)	Low	0%
	Normal	78.6%
	High	21.4%
BMI (HA Asian standard)	Underweight	3.3%
	Normal weight	31.0%
	Pre-obesity	23.8%
	Obesity class (mid)	32.4%
	Obesity class [18]	9.5%
Blood Glucose (WHO HK standard)	Low	10.5%
	Normal	83.8%
	High	5.7%
Drinker	No	93.7%
	Yes	6.3%
Smoker	No	92.6%
	Yes	7.4%
Pain Position: Elbow	No	95.7%
	Yes	4.3%
Pain Position: Foot	No	79.5%
	Yes	20.5%
Pain Position: Hand	No	70.5%
	Yes	29.5%
Pain Position: Knee	No	63.8%
	Yes	36.2%
Pain Position: Lower back	No	57.6%
	Yes	42.4%
Pain Position: Neck	No	79.5%
	Yes	20.5%
Pain Position: Shoulder	No	67.6%
	Yes	32.4%
Pain Position: Upper back	No	92.4%
	Yes	7.6%
Pain Position: Other	No	91.9%
	Yes	8.1%
Pain intensity: mild condition	Mean (SD)	3.0 (2.4)
Pain intensity: severe condition	Mean (SD)	6.5 (1.9)
Residential Population Density	Mean (SD)	1970.9 (731.0)
Subdivided units	No	96.2%
	Yes	3.8%
Indication of re-entrant bay	No	10.7%
	Yes	89.3%
Presence of Terrace	No	63.1%
	Yes	36.9%
Cardinal Orientation	East	8.10%
	North	11.90%
	Northeast	14.40%
	Northwest	15.70%
	South	13.10%
	Southeast	10.60%
	Southeast and Northwest	2.50%
	Southwest	12.50%
	West	11.30%
Number of light wells	Mean (SD)	12.71 (7.95)
Number of Lifts	Mean (SD)	5.02 (1.31)
Number of corridors of the building	Mean (SD)	3.79 (1.93)
Age of the building	Mean (SD)	26.04 (9.66)
Number of flats	Mean (SD)	695.98 (238.34)
Proportion of flat with 1–2 non-functional rooms in the building (vs. 3+ non-functional rooms)		36%
Follow-up Plan assigned	No	11.4%
	Yes	88.6%
COVID period	wave 4–5	47.6%
	wave 5	52.4%

**Table 2 ijerph-21-00179-t002:** (**a**) A Feature Importance ranking and overall scores for individual-level model without built environment features. (**b**) Feature Importance ranking for model with built environment related features.

(a)						
Description	Mood	Daily Activities	Sleeping Quality	Social Life	Work Performance	Score
Age	1	1	1	1	2	99
Main Occupation	4	2	3	4	1	91
Pain intensity: mild condition	2	3	2	3	4	91
Pain intensity: severe condition	3	4	4	2	3	89
BMI	5	5	5	5	5	80
Blood Glucose	7	6	9	12	6	65
Pain Position: Shoulder	8	9	8	6	10	64
Pain Position: Lower back	9	7	7	8	11	63
Pain Position: Knee	10	8	10	10	8	59
Pain Position: Hand	12	11	6	9	9	58
Pain Position: Neck	6	13	11	13	12	50
Blood Pressure	14	12	13	11	7	48
Pain Position: Foot	13	14	12	7	14	45
Sex	11	10	15	14	13	42
Pain Position: Other	15	16	16	16	15	27
Pain Position: Upper back	18	15	14		16	21
Drinker	16		18	15	17	18
Smoker		17	17		18	11
Pain Position: Elbow	17			17		8
Sub-divided flat		18		18		6
**(b)**						
**Description**	**Mood**	**Daily Activities**	**Sleeping Quality**	**Social Life**	**Work Performance**	**Score**
Cardinal orientation	1	1	4	1	1	97
Age	3	2	1	2	3	94
Main Occupation	5	3	3	7	2	85
Pain intensity: mild condition	2	4	5	4	7	83
Pain intensity: severe condition	4	8	2	5	6	80
Proportion of flat with <3 non-functional room	7	7	6	3	4	78
Age of the building	6	5	7	6	8	73
BMI	8	6	8	8	5	70
Number of corridors of the building	11	11	10	13	9	51
Number of lifts	12	12	13	15	11	42
Number of light wells	9	9		9		36
Pain Position: Lower back	14	14	14	10	14	39
Pain Position: Knee	13	10	11	17		33
Blood Pressure	15	15	16	14	12	33
Pain Position: Hand	16	17	9	18	16	29
Pain Position: Neck	10	16	15		17	26
Blood Glucose	17		12		10	24
Presence of terrace	18			12	15	18
Pain Position: Shoulder		18		11		13
Pain Position: Foot	19		17	19	13	16
Pain Position: Other		13				8
Pain Position: Upper back				16		5
Sex					18	3

**Table 3 ijerph-21-00179-t003:** Weights assigned to features selected by individual-level vs. individual-and-building-level models.

	Model without Built Environment Related Features			Model with Built Environment Related Features		
	(AUC: 0.7129)			(AUC: 0.9085)		
Feature	Feature Level	Rank	Score	Feature Level	Rank	Score
Age	<36	1	3	<34.2	6	5
	[36, 44)		3	[34.2, 42)		3
	[44, 67)		6	[42, 66.4)		3
	[67, 73)		6	[66.4, 70)		0
	≥73		0	≥70		0
Occupation	Catering	2	11	Catering	5	6
	Cleaner		9	Cleaner		8
	Clerk		12	Clerk		6
	Driver		4	Driver		0
	Housewife		10	Domestic Helper		6
	Sales		11	Sales		4
	Security		0	Security		4
	Technician		2	Technician		3
	Construction/manual work		8	Construction/manual work		9
Pain intensity: mild condition	<5	3	0	<1	2	0
	[5, 7.55)		5	[1, 5)		2
	≥7.55		12	[5, 6.85)		7
				≥6.85		7
Pain intensity: severe condition	<3	4	3	<4	4	0
	[3, 5)		3	[4, 5)		0
	[5, 8)		9	[5, 8)		3
	[8, 9.55)		9	[8, 9)		3
	≥9.55		21	≥9		6
BMI	(1) Underweight	5	0	(1) Underweight	9	0
	(2) Normal weight		0	(2) Normal weight		0
	(3) Pre-obesity		9	(3) Pre-obesity		0
	(4) Obesity class (mid)		9	(4) Obesity class (mid)		1
	(5) Obesity class (severe)		9	(5) Obesity class (severe)		1
Blood Glucose	(1) Low	6	0	(1) Low		
	(2) Normal		0	(2) Normal		
	(3) Abnormal		5	(3) Abnormal		
Pain Position: Knee	No	7	0	No	14	0
	Yes		2	Yes		4
Pain Position: Lower back	No	8	0			
	Yes		3			
Pain Position: Foot	No	9	0			
	Yes		1			
Pain Position: Shoulder	No	10	0	No	15	0
	Yes		2	Yes		1
Pain Position: Hand	No	11	0	No	11	0
	Yes		2	Yes		3
Pain Position: Neck	No	12	0			
	Yes		2			
Sex	F	13	1			
	M		0			
Blood Pressure	(1) Normal	15	0	(1) Normal		
	(2) High		2	(2) High		
Pain Position: Upper back	No	17	0	No		
	Yes		4	Yes		
Cardinal Orientation				East	1	7
				North		9
				Northeast		6
				Northwest		12
				South		5
				Southeast		0
				Southeast and Northwest		7
				Southwest		6
				West		5
Proportion of flat with >2 non-functional room				<0.0326	3	7
				[0.0326, 0.125)		7
				[0.125, 0.461)		3
				[0.461, 0.709)		3
				≥0.709		0
Number of flat				<267	7	0
				[267, 564)		3
				[564, 799)		10
				[799, 1140)		10
				≥1140		10
Age of the building				<11	8	0
				[11, 17)		0
				[17, 31)		8
				[31, 37.8)		8
				≥37.8		8
# of light wells				<2.3	10	5
				[2.3, 5)		5
				[5, 16)		0
				[16, 27)		0
				≥27		0
Number of corridors of the building				<2	12	4
				[2, 4)		10
				≥4		10
Number of lifts				<3	13	0
				[3, 4)		4
				≥4		10

**Table 4 ijerph-21-00179-t004:** (**a**) The number of pain-interfered areas across different range of total score tallied from weights assigned to features selected from an individual-level-only model. (**b**) The number of pain-interfered areas across different range of total score tallied from weights assigned to features selected from a mixed individual- and building-level model.

(a)				
Total Score	0–1 Area	2 Areas	3 Areas	4+ Areas
[21, 40]	0.600	0.250	0.075	0.075
[41, 60]	0.104	0.294	0.362	0.239
[61, 80]	0.038	0.192	0.308	0.462
[81, 100]	0.000	0.000	0.000	1.000
**(b)**				
**Total Score**	**0–1 Area**	**2 Areas**	**3 Areas**	**4+ Areas**
[31, 40]	1.000	0.000	0.000	0.000
[41, 50]	0.636	0.364	0.000	0.000
[51, 60]	0.214	0.271	0.357	0.157
[61, 100]	0.000	0.000	0.000	1.000

## Data Availability

The studied NGO owns the data, and hence we cannot share it publicly. The datasets generated and/or analyzed during the current study are not publicly available due to restrictions being put on data sharing by Hong Kong’s Personal Data (Privacy) Ordinance (Cap. 486) (PDPO), including, but not exclusive to, PDPO’s Guidance Note in Cross-border Data Transfer. In addition, the Research Ethics Committees of the Chinese University of Hong Kong did not give the PI permission to make public the data concerning recipients of any social services.

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
