# Peer review of "Impact of Environment on Pain among the Working Poor: Making Use of Random Forest-Based Stratification Tool to Study the Socioecology of Pain Interference"

_ijerph, 2024, doi:10.3390/ijerph21020179_

Round 1

Reviewer 1 Report

Comments and Suggestions for Authors

Dear Authors, 

I suggest the following:

-  Insert a description of random forest-based stratification into  the text for the benefit of reader who are unfamiliar with this  technique. This could be done at about lines 94 page 2.

-  Table 3. suggest heading of  columns  be altered to indicate difference between individual-level and individual-and- building-level models.

- Table 4.  Suggest adding a heading to Column 1 of  Tables 1 a & b. 

- Page 2.  Line 73 , word 'mold' should be 'mould'. 

- Page 4.  Line 18,  - deep learning models rather than model 

- page 13. line 385 - ' health weak'  - meaning ?

- Page 17 line 610.  why is reference 78 capitalised ?  

Author Response

Reviewer 1

Insert a description of random forest-based stratification into the text for the benefit of reader who are unfamiliar with this technique. This could be done at about lines 94 page 2.

Response: Thanks for the suggestion. A brief description of random-based stratification is included at the end of the background section and more details under section 2.3. “This current study applied a novel random forest algorithmwhich is commonly-used machine learning algorithm combining the output of multiple decision trees to reach a single result to model and quantify the unique contribution of a diverse ensemble of environmental, sociodemographic, and clinical factors to pain interference among the working poor. Details are described under section 2.3.”

Table 3. suggest heading of columns be altered to indicate difference between individual-level and individual-and- building-level models.

Response: the right panel should be ‘Model with built environment related features’

- Table 4.  Suggest adding a heading to Column 1 of Tables 4 a & b. 

Response: Column title added

- Page 2.  Line 73 , word 'mold' should be 'mould'. 

Response: Sorry for the typo. It is corrected.

- Page 4.  Line 18,  - deep learning models rather than model

Response: Thanks for alerting us the typo/ 

- page 13. line 385 - ' health weak'  - meaning ?

Response: It should be “…the associations between built-environment quality and people’s mental health”

- Page 17 line 610.  why is reference 78 capitalised ?

Response: Revision has been made and captialised is not needed.

We are most thankful for the comments from expert reviwer to improve our manuscript.

Albert Lee and Eman Leung on behalf of team

Reviewer 2 Report

Comments and Suggestions for Authors

The study is conducted using a retrospective observational design and focuses on individuals who are part of a community service program run by a non-governmental organization (NGO) in Hong Kong. The participants were categorized as individuals experiencing discomfort and residing in a state of working poverty. The predicted characteristics and results were obtained from databases and encompassed information on population characteristics, physical measurements, and pain evaluations using the Brief Pain Inventory (BPI). Thank you for the opportunity to review the work. I found it interesting and well written. I was especially impressed with the clear and thorough way that paper explained the findings related to each section and suggested explanations. This article can be included in publication after minor revisions.

1.      Kindly provide further details regarding the random forest algorithm and the data preparatory stages.

2.      Offer more precise delineations and criteria for participant selection.

3.      For greater clarity, please simplify statistical terms and incorporate visual aides.

4.      May I request clarification regarding the limitations of the study and suggestions for future research directions?

5.      Kindly elaborate on the potential applicability of your findings to distinct populations or contexts.

6.      May the authors kindly furnish additional elucidation about the plausible pragmatic implementations of the proposed paradigm within real-world contexts? Relevant literature could be added. Authors may see the following reference while revising. “Journal of Loss Prevention in the Process Industries, 2023, 85, 105166”.

7.      The content is plentiful, but some part of the reference literatures is kind of obsolete (in 5 years). Key publications should be cited as completed as possible. Please also clarify the novelty and application implication of your work in this section. I suggest authors refer to the latest literatures from “MDPI”, and other related journals. But please do not exceed 30% of all citations from sustainability. Authors may see the following reference while revising. “Safety 2023, 9(4), 84”

Author Response

Reviewer 2

Thank you for the expert advice.  We have reviswed accordingly.

  1. Kindly provide further details regarding the random forest algorithm and the data preparatory stages.

Response: A brief description of random-based stratification is included at the end of the background section and more details under section 2.3. “This current study applied a novel random forest algorithm which is commonly-used machine learning algorithm combining the output of multiple decision trees to reach a single result to model and quantify the unique contribution of a diverse ensemble of environmental, sociodemographic, and clinical factors to pain interference among the working poor.   Further elaboration of random forest algorithm is discussed under section 2.3…….

  1. 2.      Offer more precise delineations and criteria for participant selection.

Response: A more precise delineations and criteria for participant selection has been added to section 2.1

  1. For greater clarity, please simplify statistical terms and incorporate visual aides.

Response: Figure 1 and a high-level description of the statistical processes have been added to section 2.3

  1. May I request clarification regarding the limitations of the study and suggestions for future research directions?

Response: The limitations of the study and suggestions for future research directions are added at the end of the discussion.

The study subjects were recruited from one NGO only and the sample size is not very larger. The study subjects are from grass root mainly living in poor conditions which fits the criteria of working poor.  The random forest-based algorithm can select features by integrating individual decision trees’ rank-ordering of features based on each feature’s unique and combined contribution to the study outcomes, and optimize the cutoff values defining the response levels of selected features assigning the weights to the corresponding response levels of the selected features. The method can accommodate the current sample size.

It would further enrich the findings if future study can recruit members from local District Health Centre (a hub for primary healthcare services) and the Centre would conduct pain assessment into different categories of severity.  Similar studies would be conducted in other districts including those districts with better living conditions and higher socio-economic status.  This would enable more detail comparative analysis of impact of environment on pain on people of different socio-economic status.

The findings were discussed in the context of their benefit of informing community pain screening to target residential areas whose built environment contributed most to pain interference and informing the design of intervention programs to minimize pain interference among those who suffered from chronic pain and showed specific characteristics.

  1. Kindly elaborate on the potential applicability of your findings to distinct populations or contexts.

Response: The findings were discussed in the context of their benefit of informing community screening to target residential areas whose built environment contributed most to pain interference and informing the design of intervention programs to minimize pain interference among those who suffered from chronic pain and showed specific characteristics. This is particularly important for the lower socio-economic status as pain interference has greater impact on their livelihood.  This further elaborated in conclusion section.

  1. May the authors kindly furnish additional elucidation about the plausible pragmatic implementations of the proposed paradigm within real-world contexts? Relevant literature could be added. Authors may see the following reference while revising. “Journal of Loss Prevention in the Process Industries, 2023, 85, 105166”.

Response: We are indeed enlightened by the suggestion and has added literature to highlight the usefulness of random forest algorithm in investigation of important variable for health outcomes.  The current study is in alignment with a small but emerging body of literature that classifies clinical outcome by modelling the unique and combined contributions of an ensemble of comprehensive and intricately related features with random frost algorithm.  We have added in our discussion with references.

One example of using random forest algorithm is to investigate important variables for patient safety as researchers can predict accurate and stable relationships between variables, and healthcare quality knowledge, organizational factors, and top management objectives were found to play a crucial influence in determining patient safety grades. (Haripriya et al, 2020). The algorithm can automatically handle interactions with accurate prediction even when a large number of variables are present. Random Forest Algorithms has built-in cross-validation capability to enable the independent variables to be ranked according to their association with the outcome variable, from most effective to least effective. (Swarn et al, 2020). There are many variables associated with health outcomes in real world of healthcare delivery and it is not also easy to predict the relationship between variables even with advanced statistics. Random forest algorithm is useful to for investigation of risk predictors of different health outcomes in the complex healthcare environment.

Haripriya, G & Abinaya, K & Aarthi, N & Kumar, P & Darbari, Swati. (2021). Random Forest Algorithms in Health Care Sectors: A Review of Applications. International Journal of Recent Development in Computer Technology & Software Applications 2012; 5. 1-10

Swarn Avinash Kumar, Harsh Kumar, Vishal Dutt, Pooja Dixit, “The Role of MachineLearning in COVID-19 in Medical Domain: A Survey”. Journal on Recent Innovation in Cloud Computing, Virtualization & Web Applications 2020’ Vol 4 No 1.

  1. The content is plentiful, but some part of the reference literatures is kind of obsolete (in 5 years). Key publications should be cited as completed as possible. Please also clarify the novelty and application implication of your work in this section. I suggest authors refer to the latest literatures from “MDPI”, and other related journals. But please do not exceed 30% of all citations from sustainability. Authors may see the following reference while revising. “Safety 2023, 9(4), 84

Response: We are very thankful for the expert comments. We have removed some old literature and some latest literatures from “MDPI” have been added.  Notwithstanding the knowledge on built environment and health, the association between built environments and pain remains inadequately explored. How our study is aligned with the more recent literature, and what novelty does this study bring to the fold, have been discussed in the Discussion section in the context of its implication in public health application.

We are most thankful for the comments from expert reviwer to improve our manuscript.

Albert Lee and Eman Leung on behalf of team

Reviewer 3 Report

Comments and Suggestions for Authors

Based on what theory supports environmental factors affecting pain perception?

The explanation of the connection between pain, the working poor and the environment was quite brief. The relationship between socioecology (social and ecological factors) and pain interference needs to be specifically explored.

In the introduction, could you explain more about what a random forest algorithm is?

And why random forest algorithm is suitable is employed to conducting this kind of modelling and quantifying?

Whether this method is commonly used in similar types of research.

What are the limitations of this study?

Author Response

Reviewer 3

Thank you for the expert advice. We havd revised accordingly.

Based on what theory supports environmental factors affecting pain perception?

Response: Recent update by Trachsel et al has highlighted that the activation of peripheral and central sensitization pathways involveseveral mechanisms involving through the sensitization of peripheral nociceptors, and alterations in spinal dorsal horn neurons, and central nervous system (CNS) brain areas, can trigger a pathogenetic cascade that ends with the development of chronic pain. Again, pieces of evidence suggest the paramount role of the environment (i.e., epigenetic) and genetics.  This literature is added.

Trachsel LA, Munakomi S, Cascella M. Pain Theory. [Updated 2023 Apr 17]. In: StatPearls [Internet]. Treasure Island (FL): StatPearls Publishing; 2023 Jan-. Available from: https://www.ncbi.nlm.nih.gov/books/NBK545194/

The explanation of the connection between pain, the working poor and the environment was quite brief. The relationship between socioecology (social and ecological factors) and pain interference needs to be specifically explored.

Response: More latest literatures are added to explain the connection between pain, the working poor and the environment so the relationship between socioecology and pain interference is more specially explored.

In the introduction, could you explain more about what a random forest algorithm is?

Response: explanation added in the introduction and also refer the audience to section 2.3

And why random forest algorithm is suitable is employed to conducting this kind of modelling and quantifying?  Whether this method is commonly used in similar types of research.

Response: Thank you for the important question.  One example of using random forest algorithm is to investigate important variables for patient safety as researchers can predict accurate and stable relationships between variables, and healthcare quality knowledge, organizational factors, and top management objectives were found to play a crucial influence in determining patient safety grades. (Haripriya et al, 2020). The algorithm can automatically handle interactions with accurate prediction even when a large number of variables are present. Random Forest Algorithms has built-in cross-validation capability to enable the independent variables to be ranked according to their association with the outcome variable, from most effective to least effective. (Swarn et al, 2020). There are many variables associated with health outcomes in real world of healthcare delivery and it is not also easy to predict the relationship between variables even with advanced statistics. Random forest algorithm is useful to for investigation of risk predictors of different health outcomes in the complex healthcare environment.

Haripriya, G & Abinaya, K & Aarthi, N & Kumar, P & Darbari, Swati. (2021). Random Forest Algorithms in Health Care Sectors: A Review of Applications. International Journal of Recent Development in Computer Technology & Software Applications 2012; 5. 1-10

Swarn Avinash Kumar, Harsh Kumar, Vishal Dutt, Pooja Dixit, “The Role of MachineLearning in COVID-19 in Medical Domain: A Survey”. Journal on Recent Innovation in Cloud Computing, Virtualization & Web Applications 2020’ Vol 4 No 1.

What are the limitations of this study?

The study subjects were recruited from one NGO only and the sample size is not very larger. The study subjects are from grass root mainly living in poor conditions which fits the criteria of working poor.  The random forest-based algorithm can select features by integrating individual decision trees’ rank-ordering of features based on each feature’s unique and combined contribution to the study outcomes, and optimize the cutoff values defining the response levels of selected features assigning the weights to the corresponding response levels of the selected features. The method can accommodate the current sample size.

It would further enrich the findings if future study can recruit members from local District Health Centre (a hub for primary healthcare services) and the Centre would conduct pain assessment into different categories of severity.  Similar studies would be conducted in other districts including those districts with better living conditions and higher socio-economic status.  This would enable more detail comparative analysis of impact of environment on pain on people of different socio-economic status.

The findings were discussed in the context of their benefit of informing community pain screening to target residential areas whose built environment contributed most to pain interference and informing the design of intervention programs to minimize pain interference among those who suffered from chronic pain and showed specific characteristicson.

On behalf of the team, we are most thankful for the comments from our expert reviewer to further improve our manuscript.

Albert Lee and Eman Leung